

# Validation of XCO₂ and XCH₄ retrieved from a portable Fourier transform spectrometer with those from in-situ profiles from aircraft borne instruments

5   Hirofumi Ohyama[1], Isamu Morino[1], Voltaire A. Velazco[2,3], Theresa Klausner[4], Gerry Bagtasa[5], Matthäus Kiel[6], Matthias Frey[1], Akihiro Hori[1], Osamu Uchino[1], Tsuneo Matsunaga[1], Nicholas Deutscher[2], Joshua P. DiGangi[7], Yonghoon Choi[7], Glenn S. Diskin[7], Sally E. Pusede[8], Alina Fiehn[4], Anke Roiger[4], Michael Lichtenstern[4], Hans Schlager[4], Pao K. Wang[9], Charles C.-K. Cho[9], Maria Dolores Andrés-Hernández[10], and John P. Burrows[10]

[1] National Institute for Environmental Studies, Tsukuba, Japan

[2] Centre for Atmospheric Chemistry, University of Wollongong, NSW 2522, Australia

[3] Oscar M. Lopez Center for Climate Change Adaptation and Disaster Risk Mgmt. Foundation Inc., Manila, Philippines

15   [4] Deutsches Zentrum für Luft- und Raumfahrt (DLR), Institut für Physik der Atmosphäre, Oberpfaffenhofen, Germany

[5] Institute of Environmental Science & Meteorology, University of the Philippines, Diliman Quezon City, Philippines

[6] NASA Jet Propulsion Laboratory, California Institute of Technology, Pasadena, CA, USA

20   [7] NASA Langley Research Center, Hampton, VA, USA

[8] Department of Environmental Sciences, University of Virginia, Charlottesville, VA, USA

[9] Research Center for Environmental Changes Academia Sinica, Taipei, Taiwan

[10] Institute of Environmental Physics, University of Bremen, Otto-Hahn-Allee 1, 28359 Bremen, Germany





## Abstract

Column-averaged dry-air mole fractions of carbon dioxide ($XCO_2$) and methane ($XCH_4$) measured by a solar viewing portable Fourier transform spectrometer (FTS, EM27/SUN) have been characterized and validated by comparison using in-situ profile measurements made during the transfer flights of two aircraft campaigns: Korea-United States Air Quality Study (KORUS-AQ) and Effect of Megacities on the Transport and Transformation of Pollutants on the Regional and Global Scale (EMeRGe). The aircraft flew over two Total Carbon Column Observing Network (TCCON) sites: Rikubetsu, Japan (43.46° N, 143.77° E) for the KORUS-AQ campaign and Burgos, Philippines (18.53° N, 120.65° E) for the EMeRGe campaign. The EM27/SUN was deployed at the corresponding TCCON sites during the overflights. The mole fraction profiles obtained by the aircraft over Rikubetsu differed between the ascending and the descending flights above approximately 8 km for both $CO_2$ and $CH_4$. Because the spatial pattern of tropopause heights based on potential vorticity values from the ERA5 reanalysis show that the tropopause height over the Rikubetsu site was consistent with the descending profile, we used only the descending profile to compare with the EM27/SUN data. Both the $XCO_2$ and $XCH_4$ derived from the descending profiles over Burgos were lower than those from the ascending profiles. Output from the Weather Research and Forecast Model indicate that higher $CO_2$ for the ascending profile originated in central Luzon, an industrialized and densely populated region about 400 km south of the Burgos TCCON site. Air masses observed with the EM27/SUN overlap better with those from the descending aircraft profiles than those from the ascending aircraft profiles with respect to their properties such as origin and atmospheric residence times. Consequently, the descending aircraft profiles were used for the comparison with the EM27/SUN data. The EM27/SUN $XCO_2$ and $XCH_4$ data were derived by using the GGG2014 software in which air mass independent correction factors utilized for the TCCON data (0.9898 for $XCO_2$ and 0.9765 for $XCH_4$) were not applied. The comparison of the EM27/SUN observations with the aircraft data revealed that on average, the EM27/SUN $XCO_2$ data were biased low by 1.22 % and the EM27/SUN $XCH_4$ data were biased low by 1.67 %. The resulting air mass independent correction factors of 0.9878 for $XCO_2$ and 0.9833 for $XCH_4$ were obtained for the portable FTS.



## 1. Introduction

Greenhouse gas (GHG) total column abundances are retrieved from ground-based high-resolution Fourier transform spectrometers (FTSs) that record solar absorption spectra in the near-infrared spectral region. Presently, there are more than twenty-five such FTS observation sites across the globe forming the Total Carbon Column Observing Network (TCCON) (Wunch et al., 2011a). Stringent conditions placed on instrumentation, measurement procedures, and data processing, as well as validation to the World Meteorological Organization's (WMO) standards by comparison with aircraft and AirCore profile data (Deutscher et al., 2010; Wunch et al., 2010; Messerschmidt et al., 2011; Geibel et al., 2012; Sha et al., 2019) facilitate highly accurate and precise measurements of column-averaged dry-air mole fractions of $CO_2$ and $CH_4$ ($XCO_2$ and $XCH_4$) ($2\sigma$ uncertainties: 0.8 ppm for $XCO_2$ and 7 ppb for $XCH_4$). The TCCON data are used extensively for carbon cycle studies and play a vital role in validating space-borne data from the Greenhouse Gases Observing Satellite (Yoshida et al., 2013), the Orbiting Carbon Observatory-2 (O'Dell et al., 2018; Kiel et al., 2019), the TanSat (Liu et al., 2018), the Scanning Imaging Absorption Spectrometer for Atmospheric Chartography (Dils et al., 2014), and the Tropospheric Monitoring Instrument (Hu et al., 2018).

The Bruker IFS 125HR is at present the most stable high-resolution FTS commercially available and is currently the primary instrument selected for use at TCCON sites. However, it is expensive and its operation and maintenance requires a large infrastructure and an experienced specialist. Within the last decade, a portable and robust FTS (Bruker EM27/SUN) was developed for GHG column measurements (Gisi et al., 2012). The EM27/SUN was mainly used in observation campaigns for the quantification of local sources and sinks of GHGs. To date, citywide campaigns were conducted in urban areas such as Berlin (Hase et al., 2015), Los Angeles (Chen et al., 2016), Paris (Vogel et al., 2019), and Tokyo (Frey et al., 2017). An additional observation campaign for satellite data validation was conducted in the desert areas of Australia (Velazco et al., 2019). Long-term observations have also been conducted in Africa where operational observation by the IFS 125HR is difficult (Frey et al., 2020).

To validate EM27/SUN data, Frey et al. (2019) compared individual EM27/SUN instruments that are located around the world with a reference EM27/SUN instrument. The reference data were scaled to be consistent with a collocated IFS 125HR in Karlsruhe, Germany (Kiel et al., 2016), and empirical correction factors for each instrument were determined for $XCO_2$ and $XCH_4$ data. In March 2016 our (National Institute for Environmental Studies: NIES) EM27/SUN was delivered with a single channel for $CO_2$ and



$CH_4$ observations. In December 2017, it was sent to Bruker Optics, Inc. to add a second channel for carbon monoxide (CO) observations. A comparison with the reference EM27/SUN with both instruments operating side by side was attempted at the Karlsruhe Institute of Technology. However, consecutive periods of poor weather conditions prevented the intercomparison. In the present study, we independently validated the retrieved data products from our instrument using campaign-based aircraft measurements.

We obtained in-situ aircraft profiles of $CO_2$ and $CH_4$ over two TCCON sites (Rikubetsu, Japan (43.46° N, 143.77° E, 380 m a.s.l., Morino et al., 2018c) and Burgos, Philippines (18.53° N, 120.65° E, 35 m a.s.l., Velazco et al., 2017; Morino et al., 2018b)) in the track of the transfer flights of two aircraft campaigns: the Korea-United States Air Quality Study (KORUS-AQ); and the Effect of Megacities on the Transport and Transformation of Pollutants on the Regional and Global Scale (EMeRGe). Although the primary objectives of the overflights were to validate the TCCON $XCO_2$ and $XCH_4$ data, we also deployed our EM27/SUN at the TCCON sites during the overflights to validate the EM27/SUN data and to inter-compare between the EM27/SUN and TCCON data. In this paper, we primarily focus on the validation of the EM27/SUN data by comparison with the aircraft measurements.

## 2. Data

### 2.1 EM27/SUN

The EM27/SUN measures $XCO_2$ and $XCH_4$ values with high accuracy and precision based on solar absorption measurements (Gisi et al., 2012). The EM27/SUN features a pendulum interferometer with two corner cube mirrors and a $CaF_2$ beam splitter and has a spectral resolution of $0.5$ cm$^{-1}$ ($1.8$ cm of optical path difference); a 127 mm parabolic mirror together with the 0.6 mm aperture defines a semi field of view (FOV) of 2.36 mrad, corresponding to an external FOV of approximately 50% of the apparent solar disc diameter.

In March 2016 we started making solar absorption measurements in Tsukuba, Japan (36.05° N, 140.12° E; 31 m a.s.l.), using an EM27/SUN equipped with a standard indium gallium arsenide (InGaAs) detector covering the spectral range of 5500–11000 cm$^{-1}$ operated at ambient temperature. In December 2017, the second channel with an extended InGaAs detector element and a wedged germanium filter to limit the spectral range to 4000–5500 cm$^{-1}$ were added to enable CO measurements (Hase et al., 2016). One measurement consisted of 10 double-sided interferograms (5 interferograms each for forward and backward scans), which were separately integrated and recorded in DC mode with a sampling rate of 10 kHz; each measurement took approximately 60 s to complete.





The open-source software package GGG2014 was used for data processing and analysis (Wunch et al., 2015). The spectra were computed from the raw interferograms by applying a

fast Fourier transform. In the course of processing, any solar intensity variations that occurred during an interferogram acquisition as well as phase errors were corrected. The central algorithm of the data processing, the GFIT nonlinear least-squares fitting algorithm, scales an a priori profile to make the best spectral fit, and the retrieved column abundance was computed as the product of the a priori column abundance and the derived scaling factor. The

retrieved column abundances were then converted to column-averaged dry-air mole fractions by dividing them by the dry-air columns that were computed by retrieving the $O_2$ column abundances from the same spectra. Although the solar intensity variations were corrected, only the retrieved data with a solar intensity variation less than 1 % were used for the comparisons with the aircraft data. Although the GGG2014 software includes air mass

independent correction factors for the TCCON data retrieved from IFS 125HR spectra, these values were not utilized (i.e., set to one) for the analysis of the EM27/SUN data because we separately determine the air mass independent correction factors for EM27/SUN in this study.

## 2.2 Aircraft campaigns

The KORUS-AQ campaign is an international, multi-organization mission to observe air quality across the Korean peninsula and surrounding seas from various platforms such as aircraft, ground sites, ships, and satellites. On 26 April 2016, the aircraft took off from the U.S. bound for scientific observations around Korea, which began on 1 May 2016. On its transfer flight to Korea, a dedicated maneuver over Rikubetsu was performed. In-situ measurements

of $CO_2$ and $CH_4$ over the Rikubetsu TCCON site during the KORUS-AQ campaign were performed by two instruments onboard the DC-8 aircraft: the Atmospheric Vertical Observations of $CO_2$ in the Earth's Troposphere (AVOCET) instrument using a non-dispersive infrared spectrometer (LI-COR, Inc. LI-6252) for $CO_2$ and the Differential Absorption Carbon monOxide Measurement (DACOM) instrument based on infrared

wavelength modulation spectroscopy for $CH_4$. Calibrations of both instruments were performed during flight using standard gases traceable to the WMO scale. The sampling rates for both measurements were 1 Hz. Additional radiosonde observations (Meisei Electric Co., Ltd. RS-11G) were performed by the Japan Weather Association under a contract with the NIES to obtain pressure, temperature, and humidity profiles coincident with the aircraft $CO_2$

and $CH_4$ profiles.

The objective of the EMeRGe project is to investigate the impact of emissions from major population centers on air pollution at local, regional and hemispheric scales by conducting





dedicated airborne measurement campaigns. The campaigns in Europe and Asia using the High Altitude and Long Range Research Aircraft (HALO) platform were performed during the summer of 2017 (Europe) and the spring of 2018 (Asia). HALO flew over the Burgos TCCON site in the track of the transfer flight from Thailand through Manila to Tainan on 12 March 2018. In-situ $CO_2$ and $CH_4$ profiles, calibrated using standards traceable to the WMO scales, were measured with a cavity-ringdown spectrometer (CRDS, Picarro, Inc. G1301-m) onboard HALO. Ancillary data was provided by the basic meteorological sensor package that measures pressure, temperature, and humidity.

## 3. Results and Discussion

### 3.1 EM27/SUN and aircraft measurements in Rikubetsu

The EM27/SUN measurements in Rikubetsu were made from the roof of the building that houses the Rikubetsu TCCON FTS on 27 April 2016. Surface meteorological data (pressure, temperature, humidity, and wind) measured by meteorological instruments deployed as a part of the TCCON station were used for analyses of the EM27/SUN data. Figure 1a shows the flight track over Hokkaido, Japan, between 01:25 and 02:30 UTC on 27 April 2016. The descending profile was measured from 10.8 to 0.10 km with a spiral flight pattern over the Rikubetsu site. The ascending profile was measured up to an altitude of 11.5 km in a linear manner on the west side of the Rikubetsu site. The descending and ascending profiles of both $CO_2$ and $CH_4$ (Figs. 1b and 1c) were consistent with each other up to an altitude of ~8 km. The mole fractions at higher altitudes were likely affected by an intrusion of stratospheric air, which reached approximately 8 km for the descending profile and approximately 10 km for the ascending profile, as described in more detail below. Consequently, we calculated $XCO_2$ and $XCH_4$ separately for ascending and descending aircraft profiles. Each profile was averaged per layer with a layer width of 0.05 km.

We examined the causes of the differences between the descending and ascending profiles in order to determine which profiles should be used for the comparison with the EM27/SUN. For the aircraft data, the potential vorticity, which has been previously used as an indicator to determine the tropopause height (Trickl et al., 2011), was investigated along the aircraft tracks. The potential vorticity was calculated from the European Centre for Medium-Range Weather Forecasts (ECMWF) fifth generation reanalyses (ERA5) with a spatial resolution of $0.25° \times 0.25°$ and a temporal resolution of 1 h (C3S, 2017). Figure 2a shows the profiles obtained from the aircraft borne measurements above 7 km over Rikubetsu, color-coded by the corresponding potential vorticity values. We found that when the potential vorticity was



greater than approximately 3 PVU (potential vorticity units; 1 PVU = $10^{-6}$ $m^2$ $s^{-1}$ K $kg^{-1}$), the $CO_2$ and $CH_4$ mole fractions began to decrease. We, therefore, assumed that the air masses with potential vorticity values of more than 3 PVU were of stratospheric origin and that the tropopause height corresponded to 3 PVU. Figure 2b shows the latitude-longitude cross section of the geopotential height corresponding to the potential vorticity of 3 PVU at 02:00 UTC on 27 April 2016, and the altitude-longitude cross section of the potential vorticity averaged between 42° and 45°N is shown in Fig. S1 in the Supplement. A strip-shaped subsidence of the tropopause (tropopause fold) occurred over Hokkaido and the southern border of the tropopause fold occurred over Rikubetsu. The tropopause fold has been observed to form on the north side of the upper tropospheric jet stream (Holton et al., 1995), and this is apparent in Fig. 2b. In the northern extratropics e.g. Hokkaido, the tropopause fold most frequently occurs from April to June (Stohl et al., 2003). We compared the tropopauses based on the potential vorticity (dynamical tropopauses) with those determined by radiosonde temperature data (lapse rate tropopauses): the two types of tropopauses were spatially consistent (Table 1). The dynamical tropopause over Sapporo was higher than those over Rikubetsu and Wakkanai and was similar to the dynamical tropopause for the ascending profile. Because the dynamical tropopause over Rikubetsu was consistent with that of the descending profile, we decided to compare the descending profile with the EM27/SUN data.

Although the altitude range of the descending flight around Rikubetsu was limited to 0.10–10.8 km, the aircraft data covered all altitudes from the tropopause height (calculated in the GGG2014: 9.4 km) to the altitude of the ground-based instruments (elevation of the instrument: 0.38 km); consequently, there was no need to extrapolate the aircraft data in the troposphere. The aircraft data were connected to the a priori profile of EM27/SUN analysis above ceiling heights (i.e., in the stratosphere). To investigate uncertainties in the aircraft $XCO_2$ and $XCH_4$ data, we performed a sensitivity analysis in which we perturbed each source of error (i.e., measurement uncertainty and tropopause height) by a realistic amount and compared the resulting $XCO_2$ and $XCH_4$ with the corresponding unperturbed case. We separated the sources of errors into tropospheric and stratospheric parts, and the total error was estimated as a root sum square of each part. We estimated the errors in the aircraft $CO_2$ data to be 0.27 ppm on the basis of a precision of 0.1 ppm and accuracy of 0.25 ppm (Vay et al., 2011; Tang et al., 2018). The error in the stratospheric $CO_2$ mole fraction was estimated to be 0.3 %, and the perturbed $CO_2$ profile was created by shifting the a priori profile up by 1 km and adding 0.3 % error to the a priori profile (Wunch et al., 2010). For $CH_4$, the error in the aircraft data was estimated to be 0.1 % (https://www-gte.larc.nasa.gov/pem/DACOM.htm, last access: 5 September 2019). The perturbed $CH_4$ profile is created by shifting the a priori





profile up by 1 km. The estimated errors in aircraft $XCO_2$ and $XCH_4$ are listed in Table 2.

**3.2 EM27/SUN and aircraft measurements in Burgos**

The EM27/SUN was located next to a TCCON FTS container in Burgos, Ilocos Norte, Philippines during the period 7–13 March 2018. The flight track over the Philippines between 08:21 and 10:41 UTC on 12 March 2018 is shown in Fig. 3a. The descending profile was measured from 6.5 to 0.6 km approaching the Burgos site from south to northeast. The

low-level flight at approximately 0.6 km was performed as near as possible to the north side of the Burgos site. The ascending profile was measured up to 9.4 km after the low-level flight west of the Burgos site. Additional data for the profiles above 6.5 (descent flight) and 9.4 km (ascent flight) were taken from the same aircraft data measured during the descent flight from an altitude of 13.9 km west of Manila. Figures 3b and 3c show the descending and ascending

profiles of $CO_2$ and $CH_4$. Because the aircraft data were limited to 0.6–13.9 km, the aircraft data need to be extrapolated to both the surface (elevation of the EM27/SUN instrument: 0.035 km) and the tropopause height (tropopause calculated in the GGG2014: 16.6 km) using realistic assumptions. Above the ceiling altitude of the aircraft, the aircraft data in the highest layer were extrapolated to the tropopause height and then connected to the a priori profile.

Below the lowest flight altitude, the average value of aircraft data during the low-level flight near Burgos site (less than 0.55 km) were linearly extrapolated to the surface. The static pressure and temperature values and water vapor mixing ratios, recorded by airborne instruments, were used to calculate the aircraft $XCO_2$ and $XCH_4$ values. For pressure, temperature, and water vapor values below and above the aircraft altitude, we used nearby

(Laoag, Philippines) radiosonde measurements and GGG2014 a priori profile, respectively.

Compared to the profiles over Rikubetsu (Figs. 1b and 1c), the $CO_2$ and $CH_4$ mole fraction profiles obtained from the descending and ascending flights over Burgos differed substantially, notably in the lower troposphere. To explore the reasons for these differences, the spatial $CO_2$ distribution in the lower troposphere around the Burgos site was investigated

using output from Weather Research and Forecast – Chemistry (WRF-Chem) GHG tracer model (Skamarock et al., 2008) run with 5-day spin-up time (Bagtasa, 2011). The meteorological initial and boundary conditions for the simulation in this study were taken from the National Center for Environmental Prediction (NCEP) Final (FNL) Operational Model Global Tropospheric Analyses data with a spatial resolution of 1° × 1° and a temporal

resolution of 6 h (http://rda.ucar.edu, last access: 5 September 2019). The WRF-Chem Model downscales the NCEP FNL reanalysis data to a finer spatial resolution of 5 km at 3-h intervals. Figure 4a shows the simulated $CO_2$ mole fraction averaged between the surface and 3 km

altitude at 09:00 UTC on 12 March 2018. The simulation domain includes Japan, Korea, China, Taiwan, and parts of Southeast Asia including Indochina and the Philippines. The $CO_2$
emissions from fossil fuel combustion were taken from the Open-source Data Inventory for Anthropogenic $CO_2$, version 2018 (Oda and Maksyutov, 2015). Furthermore, the $CO_2$ mole fractions in the smaller region shown in Fig. 4b were simulated at 1-h intervals and 1 km resolution. Output from the WRF-Chem Model show that northeast wind was dominant on the east side of the Philippines, where there are no large emission sources. Luzon island disrupts
the northeast wind, consequently lowering wind speeds in the west of central Luzon. This disruption of wind flow possibly induced high $CO_2$ concentrations related to long residence times to the west of central Luzon. The simulated $CO_2$ concentrations below 3 km west of the Burgos site (i.e., in the ascending flight area) are a few ppm higher relative to the background (Fig. 4b), and the high $CO_2$ also seems to originate in central Luzon, an industrialized and
densely populated region about 400 km south of the Burgos TCCON site. The Burgos TCCON site is located on a wind farm and the whole province of Ilocos Norte has been designated as a "coal free" province, therefore strong point sources such as coal-fired power plants are absent in this region (Velazco et al., 2017). Because air mass properties observed with the EM27/SUN at the Burgos TCCON site are more consistent with those associated
with the descending profiles rather than the ascending profiles, the descending profiles were used for the comparison with the EM27/SUN data. Additionally, we note that the overflight time was just after sundown (approximately 10:00 UTC), and therefore the descending flight toward Burgos was closer in time to the EM27/SUN measurements.

The total measurement uncertainty in aircraft $CO_2$ data obtained with the Picarro analyzer
G1301-m was estimated to be 0.5 ppm following the calibration procedure described by Klausner et al. (2020), and the error in the $CO_2$ data extrapolated to the surface was estimated to be 1.8 ppm on the basis of standard deviations of the average values. The $CO_2$ concentrations during the low-level flight were quite variable. This behavior is attributed to local emissions and biosphere exchange. For the $CH_4$ measurements the error in aircraft data
was estimated to be 1.4 ppb, and the error in the extrapolated data was estimated to be 3.0 ppb. We estimated the contributions of the stratospheric parts to the $XCO_2$ and $XCH_4$ errors by the methods similar to the Rikubetsu cases. Table 2 lists the estimated aircraft $XCO_2$ and $XCH_4$ errors. We found that the errors in the tropospheric dry columns over Burgos were larger than those over Rikubetsu because the aircraft data over Burgos had to be extrapolated to the
surface where $CO_2$ concentrations were more variable. In contrast, the errors in the stratospheric dry columns were larger over Rikubetsu than Burgos because the tropopause height over Rikubetsu was 7.2 km lower (in the case of GGG2014 tropopause height) and,





thus, the stratospheric part larger than that over Burgos.

### 3.3 Stability of EM27/SUN measurements

To evaluate the extent of instrument drifts of the EM27/SUN due to transporting the instruments (hereafter "transports"), the instrumental line shape (ILS) of the EM27/SUN was evaluated before and after the solar absorption measurements in Rikubetsu and Burgos. We performed indoor open-path measurements of water vapor absorption lines (Frey et al., 2015) obtained in Tsukuba and analyzed the spectra utilizing the LINEFIT v14.5 software (Hase et al., 1999). The LINEFIT analysis of the data determines two ILS parameters, modulation efficiency and phase error defined by a function of optical path difference, which represent line broadening/narrowing and asymmetry, respectively. Before and after the solar absorption measurement in Rikubetsu, the modulation efficiency changed from 0.9856 to 0.9843, and the phase error changed from 0.0025 to 0.0022 rad. In the case of the transport to and from Burgos, the modulation efficiency changed from 0.9791 to 0.9847, while the phase error changed from 0.0028 to 0.0025 rad. Because a change in modulation efficiency of 0.01 induces a change in $XCO_2$ of 0.15 % (Frey et al., 2015), the change in modulation efficiency due to transport between Tsukuba and Rikubetsu/Burgos has little impact (<0.1 %) on the retrievals.

As an additional evaluation of the instrument drifts, we examined the differences from the Tsukuba TCCON data (Morino et al., 2018a) before and after the EM27/SUN transports to Rikubetsu and Burgos. The TCCON spectra are also analyzed with the GGG2014 software. We note that all the TCCON data used in the present study are scaled by air mass independent correction factors, which were derived from aircraft in-situ data in the past (Wunch et al., 2010; 2015). The retrieved $XCO_2$ and $XCH_4$ data are averaged into 10 min bins for each instrument. To compare different remote sensing data sets, the differences in the a priori profile and the column averaging kernels must be taken into account (Rodgers and Connor, 2003). The column averaging kernels represent the altitude-dependent sensitivity of the retrieved total column to the perturbation of mole fraction at a given altitude. Because the a priori profile was common for the EM27/SUN and TCCON analyses, only the difference in the column averaging kernel should be considered by adjusting the TCCON data. We denote the EM27/SUN and TCCON by subscripts 1 and 2, respectively, and the TCCON column-averaged value adjusted to the EM27/SUN column averaging kernel $a_1$, $\hat{c}_{12}$ can be expressed by the following equation (Rodgers and Connor, 2003; Wunch et al., 2011b):



$$\hat{c}_{12} = c_{\mathrm{a}} + \left( \frac{\hat{c}_2}{c_{\mathrm{a}}} - 1 \right) \sum_j h_j a_{1j} x_{\mathrm{a}j} , \tag{1}$$

where $c_{\mathrm{a}}$ is the a priori column-averaged value, $\hat{c}_2$ is the retrieved TCCON column-averaged value, $h$ is the pressure-weighting function, $x_{\mathrm{a}}$ is the a priori profile, and $j$ represents the altitude level. The overall column averaging kernels of EM27/SUN and TCCON FTS depending on solar zenith angle are shown in Hedelius et al. (2016). According to analyses using the Tsukuba TCCON data on 3 June 2016 (29 January 2018), the overall differences ($\hat{c}_{12} - \hat{c}_2$) between the adjusted TCCON value $\hat{c}_{12}$ and the original TCCON value $\hat{c}_2$ are $0.04 \pm 0.08$ ppm ($0.06 \pm 0.02$ ppm) for $XCO_2$ and $1.64 \pm 2.44$ ppb ($0.33 \pm 0.20$ ppb) for $XCH_4$. From these results, we find that the effect of the difference in column averaging kernel has little impact on the comparison between the EM27/SUN and the TCCON data, and we decided to compare the EM27/SUN data with the original TCCON data. Table 3 summarizes the differences between the EM27/SUN and TCCON data before and after the transports. Note that only the TCCON data are corrected by the air mass independent correction factors. The changes in the $XCO_2$ differences are less than 0.4 ppm for the transports to and from both Rikubetsu and Burgos, while the changes in the $XCH_4$ differences are less than 3.0 ppb. Thus, the influence of EM27/SUN transports on the $XCO_2$ and $XCH_4$ retrievals are comparable to their uncertainties.

### 3.4 Comparisons of EM27/SUN with aircraft data

To compare the EM27/SUN data with the aircraft data, the aircraft column-averaged value $\hat{c}_{\mathrm{in\,situ}}$ is calculated by considering the column averaging kernels and the a priori values of EM27/SUN analysis:

$$\hat{c}_{\mathrm{in\,situ}} = \gamma c_{\mathrm{a}} + \sum_j h_j a_{1j} (x_{\mathrm{in\,situ}} - \gamma x_{\mathrm{a}})_j , \tag{2}$$

where $x_{\mathrm{in\,situ}}$ is the in-situ aircraft profile and $\gamma$ is the scaling factor for the EM27/SUN retrieval. The EM27/SUN data recorded within $\pm 1$ h of the aircraft measurements were averaged. The EM27/SUN column averaging kernel in Equation (2) was obtained by averaging those values for multi-retrieval windows within $\pm 1$ h of the aircraft measurement. Applying the column averaging kernel to the integration of the aircraft data modifies the raw

aircraft $XCO_2$ ($XCH_4$) value by +0.15 ppm (–0.22 ppb) for the Rikubetsu overflight and +0.06 ppm (+0.35 ppb) for the Burgos overflight. We assumed the measurement time for the aircraft

to be the measurement time at the lowermost altitude. Since a common column averaging kernel is applied to the descending and ascending profiles, the differences in calculated aircraft $XCO_2$ and $XCH_4$ data between the descent and ascent flights result solely from the difference in concentrations between the two profiles.

Figure 5 shows the time series of $XCO_2$ and $XCH_4$ measured by the EM27/SUN in

Rikubetsu and Burgos. The EM27/SUN measurements taken at the overflight time were interrupted by clouds for Rikubetsu and sundown for Burgos. The numbers of EM27/SUN data, satisfying the temporal coincidence criterion, are 4 and 24 for Rikubetsu and Burgos, respectively. The aircraft $XCO_2$ and $XCH_4$ values calculated using Eq. (1) are presented separately for the descending and ascending profiles, although only the descending profiles

are used for the comparison to the EM27/SUN data as described above. The EM27/SUN column averaging kernels for the flight times over Rikubetsu and Burgos are shown in Fig. 6. Table 4 lists results of the comparison of the EM27/SUN with the aircraft $XCO_2$ and $XCH_4$ data. The relative biases of EM27/SUN $XCO_2$ with respect to the aircraft $XCO_2$ values are – 1.172 % and –1.260 % for the comparisons at Rikubetsu and Burgos, respectively. The

relative biases of EM27/SUN $XCH_4$ with respect to the aircraft $XCH_4$ values are –1.572 % and –1.772 % for the comparisons at Rikubetsu and Burgos, respectively. Overall, correction factors for EM27/SUN $XCO_2$ and $XCH_4$ values are determined to be 0.9878 and 0.9833, respectively, and corrected values are obtained by dividing the raw values by the correction factors. Errors in their correction factors were calculated from the estimated aircraft total

errors (Table 2) and EM27/SUN measurement precisions (standard deviations of the mean EM27/SUN values) and were found to be 0.0012 for $XCO_2$ and 0.0038 for $XCH_4$. The correction factors for TCCON data are 0.9898 for $XCO_2$ and 0.9765 for $XCH_4$, and the $XCH_4$ correction factor of TCCON with the higher spectral resolution (0.02 $cm^{-1}$) deviates more largely from 1 than that of EM27/SUN with the lower spectral resolution (0.5 $cm^{-1}$). Here, the

GGG2014 uses HITRAN 2008 database and a Voigt line shape to calculate absorption coefficients of $CH_4$ in the 1.67 μm band, which results in smaller $XCH_4$ for both the TCCON and EM27/SUN compared to aircraft in-situ $XCH_4$. When the spectral resolution of TCCON is reduced to that of EM27/SUN by truncating the TCCON interferogram, the retrieved low-resolution TCCON $XCH_4$ becomes consistent with the EM27/SUN $XCH_4$ (Frey et al.,

2019; Hedelius et al., 2016). This implies that the inaccurate line shape and spectroscopic parameters in the 1.67 μm band would have a larger impact on $XCH_4$ retrievals from the high-resolution spectra than those from low-resolution spectra.

Hedelius et al. (2016) compared the four EM27/SUN data with the Lamont (U.S.) TCCON data. The EM27/SUN $XCO_2$ and $XCH_4$ data had mean biases of 0.03% and 0.75% relative to the TCCON data, respectively, and the correction factors for EM27/SUN were estimated to be $0.9901 \pm 0.0011$ and $0.9839 \pm 0.0027$. Our results were in agreement with the results from Hedelius et al. (2016) within the range of the uncertainties in correction factors for TCCON data.

We compared the TCCON data with the same aircraft data as used for validating the EM27/SUN data (Fig. 5). For comparisons of the TCCON data with the aircraft data the temporal range to calculate mean TCCON values was expanded to within $\pm 2$ h, because there were few TCCON data available within $\pm 1$ h of the aircraft overpass. We note that the column averaging kernels and scaling factors in Equation (1), to calculate comparable aircraft $XCO_2$ and $XCH_4$ values, were altered to correspond to the TCCON data (Fig. 6). The comparison between the TCCON and aircraft data (Table 5) revealed that the Rikubetsu TCCON data were biased high by 0.365% for $XCO_2$ and 0.271% for $XCH_4$ while the Burgos TCCON data were in good agreement with the aircraft data, with mean differences of <0.1% for both $XCO_2$ and $XCH_4$.

In the present study, the comparisons of the EM27/SUN data with the Tsukuba TCCON data (Sect. 3.3) were restricted to the periods before and after the transports of the EM27/SUN instrument. In addition, the EM27/SUN measurements in Burgos were conducted for a week, although the results on only the overflight day were shown here because the focus of this study is the validation of the EM27/SUN data. The collocated measurements by our EM27/SUN and TCCON FTS were also performed at the TCCON site in Saga, Japan, in addition to the Tsukuba, Rikubetsu, and Burgos TCCON sites. An evaluation of the consistency between these TCCON data sets based on comparison to the EM27/SUN data will be performed in a future study.

## 4. Conclusions

The $XCO_2$ and $XCH_4$ values from an EM27/SUN have been validated by comparison with in-situ aircraft data obtained over the Rikubetsu and Burgos TCCON sites in the track of the transfer flights of the KORUS-AQ and EMeRGe campaigns, respectively. The impacts of transport on the EM27/SUN were investigated and evaluated by examining both the ILS and the differences of the $XCO_2$ and $XCH_4$ data products to those of the Tsukuba TCCON data before and after transport. We find that the influence of EM27/SUN transports on the $XCO_2$ and $XCH_4$ retrievals were comparable to their uncertainties. The aircraft profiles obtained





over the two TCCON sites vary between the descending and ascending flights. Investigation of the dynamical tropopause using the ERA5 potential vorticity values reveals that a

440 tropopause fold occurred over Rikubetsu during the measurements made at the location of the descending flight, but not during the ascending flight. The output from the WRF-Chem GHG tracer model indicates that during the ascending flight close to Burgos of the HALO, the aircraft encountered air masses having high $CO_2$, probably resulting from central Luzon. Air masses observed with the EM27/SUN were different to those encountered by HALO during

the ascending profiles. However, during the descending profiles made by HALO, the EM27/SUN measured air masses that have a similar history to those measured by HALO. On the basis of the comparison between the EM27/SUN data and the selected (descending) aircraft data, the correction factors for EM27/SUN are determined to be 0.9878 for $XCO_2$ and 0.9833 for $XCH_4$. These values are consistent with those derived from the relative differences

between EM27/SUN and TCCON data that were examined in the previous study (Hedelius et al., 2016). The comparison between the TCCON and aircraft data showed that the Rikubetsu TCCON data were biased high by 0.365% for $XCO_2$ and 0.271% for $XCH_4$ while the Burgos TCCON data and aircraft data agreed to within 0.1% for both $XCO_2$ and $XCH_4$.

**Competing interests.**

The authors declare that they have no conflict of interest.

**Acknowledgements.**

The KORUS-AQ aircraft data were obtained from the NASA Langley Research Center Airborne Science Data for Atmospheric Composition (https://www-air.larc.nasa.gov/cgi-bin/ArcView/korusaq). The ERA5 reanalyses data were acquired from the Copernicus Climate Change Service Climate Data Store (https://cds.climate.copernicus.eu/#!/home). The NCEP FNL Operational Model Global

Tropospheric Analyses data were obtained from the Research Data Archive at the National Center for Atmospheric Research, Computational and Information Systems Laboratory (http://rda.ucar.edu). TCCON data were obtained from the TCCON Data Archive, hosted by CaltechDATA, California Institute of Technology (https://tccondata.org/). We are grateful to Prof. D. W. T. Griffith for his useful comments and discussions. We thank Mr. T. Nakatsuru

for the EM27/SUN measurement at the Rikubetsu site. Local technical support for the EM27/SUN and TCCON measurements in Burgos is provided by the Energy Development Corporation (EDC, Philippines). The EM27/SUN measurements at the Rikubetsu and Burgos



TCCON sites and operations of the Rikubetsu, Tsukuba, and Burgos TCCON sites are financially supported in part by the GOSAT series project. Travel and financial support to V. A. V. were granted by NIES, UOW SMAH-PEPA, UIC International Links Grant Scheme and UOW-CAC cluster. V. A. V. thanks the Civil Aviation Authority of the Philippines, R. Mina, R. A. Casim, R. Macatangay, M. Cayetano and Dragon Air Aviation for assistance. V. A. V. thanks the First Philippine Holdings and the municipality of Burgos, Ilocos Norte. V. A. V. extends special thanks to O. M. Lopez, F. Lopez, R. Tantoco, A. de Jesus, A. Durog, C. Aguilar and all EBWPC, EDC and FPH executives and staff. The HALO (High Altitude LOng Duration) aircraft is a German government research aircraft operated for the German research community by the DLR (German Aerospace) from Oberpfaffenhofen. The Effect of Megacities on the Transport and Transformation of Pollutants on the Regional and Global Scale (EMeRGe) is a research mission selected by the German research foundation (DFG) for its HALO SPP 1294 infrastructure research programme. The flight costs of EMeRGe campaign were funded by a consortium comprising the DFG, which supports the German university costs, the Research Center for Environmental Changes Academia Sinica, Taiwan, the DLR Institute of Atmospheric Physics, DLR-IAP, the Karlsruhe Institute of Technology, KIT and Research Centre Jülich, FZ-J. The EMeRGe research undertaken at the University of Bremen and the DLR-IAP for EMeRGe was funded primarily by the University of Bremen and DLR respectively and in small part by the DFG. The University of Bremen also thanks the Max Planck Institute in Mainz for support for EMeRGe.



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





Table 1. Summary of radiosonde observations in Hokkaido, Japan, on 27 April 2016.  The last 2 columns show the lapse rate tropopause and the dynamical tropopause.

| Launch location | Latitude [°N] | Longitude [°E] | Elevation [m] | Launch time [UTC] | Lapse rate tropopause [km] | Dynamical tropopause at 3 PVU [km] |
|---|---|---|---|---|---|---|
| | | | | 00:59 | 9.17 | 8.56 |
| Rikubetsu | 43.46 | 143.77 | 370 | 03:00 | 9.47 | 8.43 |
| | | | | 04:34 | 11.07 | 8.76 |
| Sapporo | 43.05 | 141.33 | 26 | 00:00 | 10.86 | 9.66 |
| Wakkanai | 45.41 | 141.68 | 11 | 00:00 | 8.96 | 7.68 |





Table 2. Errors in aircraft $XCO_2$ and $XCH_4$ data.

| Location | $XCO_2$ errors [ppm] | | | $XCH_4$ errors [ppb] | | |
|---|---|---|---|---|---|---|
| (Campaign) | Troposphere | Stratosphere | Total | Troposphere | Stratosphere | Total |
| Rikubetsu (KORUS-AQ) | 0.22 | 0.36 | 0.42 | 1.5 | 9.6 | 9.7 |
| Burgos (EMeRGe) | 0.58 | 0.04 | 0.58 | 1.4 | 1.7 | 2.2 |


Table 3. The differences in $XCO_2$ and $XCH_4$ between the EM27/SUN and Tsukuba TCCON data (EM27/SUN minus TCCON) before and after the transports of the EM27/SUN instrument. We note that correction factors are applied to only TCCON data.

| Date | $XCO_2$ difference [ppm] | $XCH_4$ difference [ppb] |
|---|---|---|
| 11, 12, 15, 19, and 20 Apr 2016 (before Rikubetsu obs.) | $-3.86 \pm 0.48$ | $-27.1 \pm 2.0$ |
| 3, 10, and 14 Jun 2016 (after Rikubetsu obs.) | $-3.98 \pm 0.60$ | $-25.8 \pm 3.2$ |
| 29 Jan 2018 (before Burgos obs.) | $-4.24 \pm 0.58$ | $-34.2 \pm 2.0$ |
| 9, 10, 12, 13, 19, and 20 Apr 2018 (after Burgos obs.) | $-4.64 \pm 0.30$ | $-31.2 \pm 2.4$ |



Table 4. Comparison of EM27/SUN data with aircraft $XCO_2$ and $XCH_4$ data. The relative differences are calculated as follows: (EM27/SUN – Aircraft) / Aircraft × 100.

| Location (Campaign) | $XCO_2$ | | | $XCH_4$ | | |
|---|---|---|---|---|---|---|
| | EM27/SUN [ppm] | Aircraft [ppm] | Relative difference [%] | EM27/SUN [ppb] | Aircraft [ppb] | Relative difference [%] |
| Rikubetsu (KORUS-AQ) | 400.49 | 405.24 | −1.172 | 1784.8 | 1813.3 | −1.572 |
| Burgos (EMeRGe) | 402.64 | 407.78 | −1.260 | 1823.4 | 1856.3 | −1.772 |






Table 5. Comparison of TCCON data with aircraft $XCO_2$ and $XCH_4$ data. The relative differences are calculated as follows: (TCCON – Aircraft) / Aircraft × 100.

| Location (Campaign) | $XCO_2$ | | | $XCH_4$ | | |
|---|---|---|---|---|---|---|
| | TCCON [ppm] | Aircraft [ppm] | Relative difference [%] | TCCON [ppb] | Aircraft [ppb] | Relative difference [%] |
| Rikubetsu (KORUS-AQ) | 406.45 | 404.97 | 0.365 | 1814.7 | 1809.8 | 0.271 |
| Burgos (EMeRGe) | 407.53 | 407.78 | −0.061 | 1855.5 | 1855.4 | 0.005 |




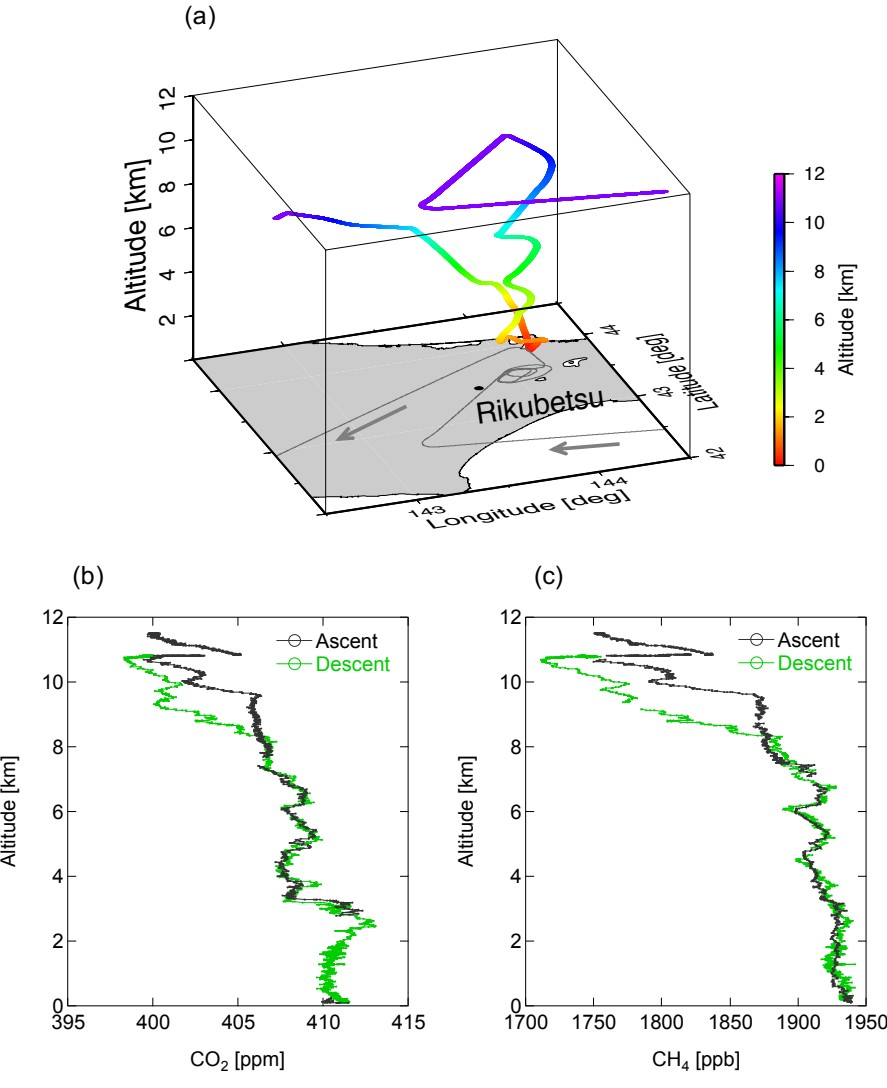

Figure 1. (a) Aircraft flight track over Hokkaido, Japan, on 27 April 2016 during the KORUS-AQ campaign. The arrows indicate the flight direction and the thin solid line represents the flight track projected on the ground. (b, c) The descending (green) and ascending (black) $CO_2$ and $CH_4$ mole fraction profiles that are used for calculating the column-averaged dry-air mole fractions.


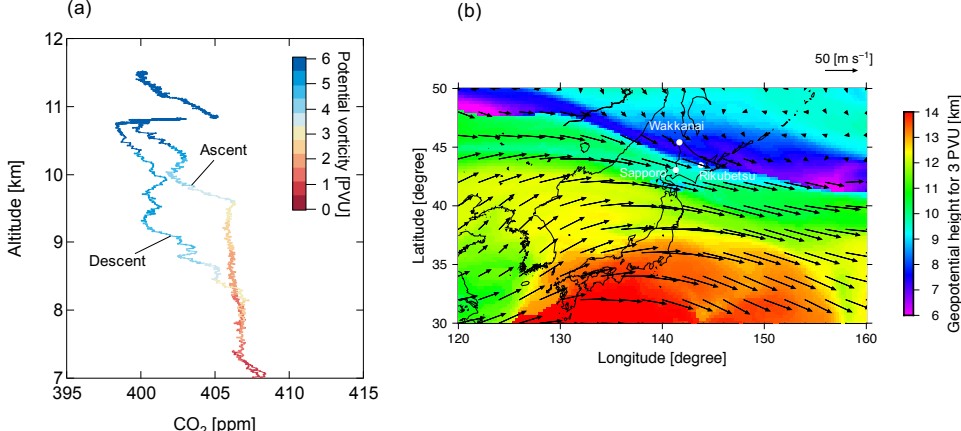

Figure 2. (a) The $CO_2$ profiles above 7 km over Rikubetsu. Colors denote the potential

vorticity values from the ERA5 (see text for details). (b) The ERA5 geopotential height (color

scale) and winds (vectors) at the 3 PVU level on 27 April 2016, 02:00 UTC are shown. White

dots indicate the locations of Rikubetsu and two radiosonde stations in Hokkaido operated by

the Japan Meteorological Agency (Sapporo and Wakkanai).



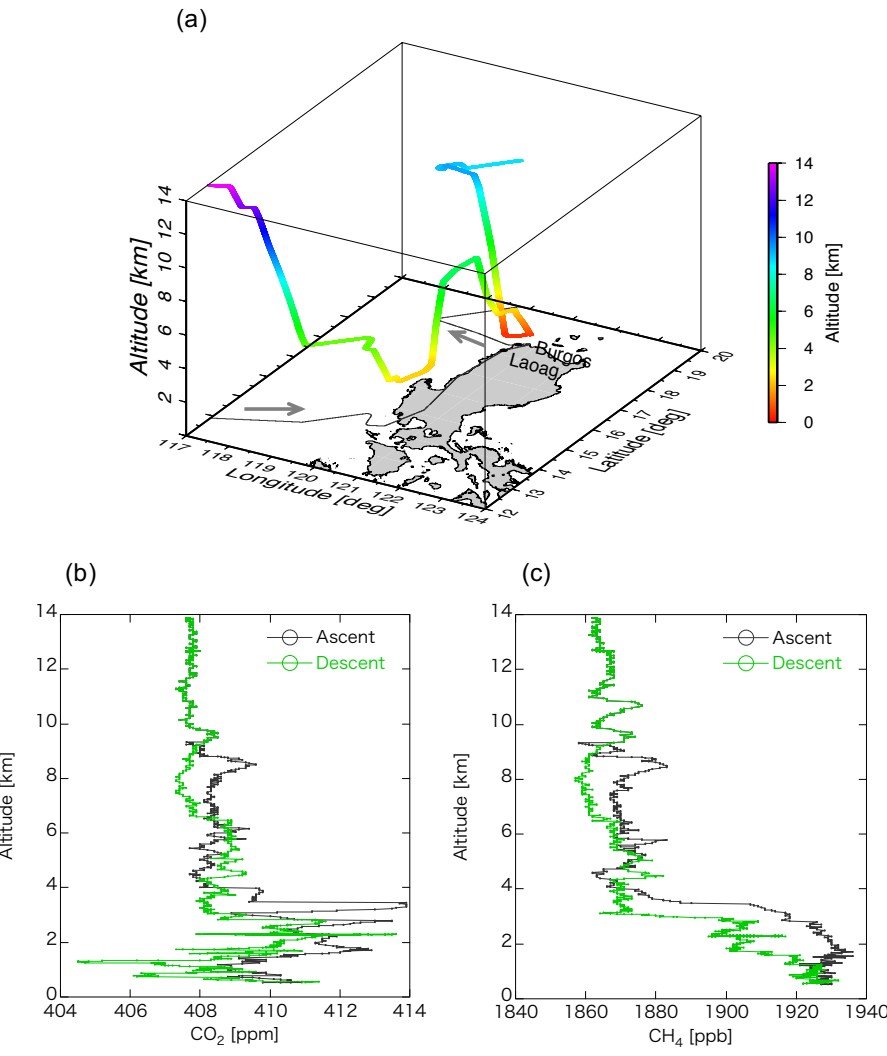


Figure 3. (a) Aircraft flight track over the Philippines on 12 March 2018 during the EMeRGe campaign. The arrows indicate the flight direction and the thin solid line represents the flight track projected on the ground. (b, c) The descending (green) and ascending (black) $CO_2$ and $CH_4$ mole fraction profiles used for calculating their column-averaged dry-air mole fractions

are shown.



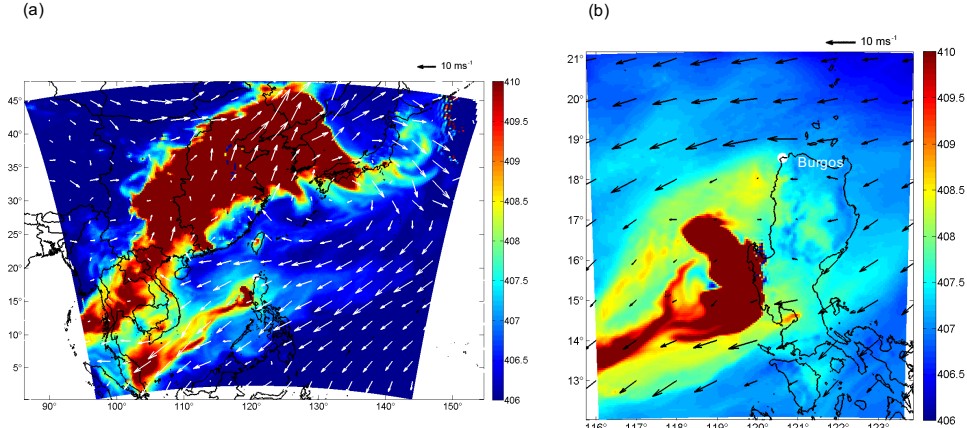

Figure 4. (a) Mean $CO_2$ mole fractions and wind vectors from the surface to 3 km altitude over Japan, Korea, China, Taiwan, and parts of Southeast Asia at 09:00 UTC on 12 March 775 2018, simulated by the Weather Research and Forecast Model. (b) Same as Figure 4a, but for a magnified section over the Philippines at 10:00 UTC.

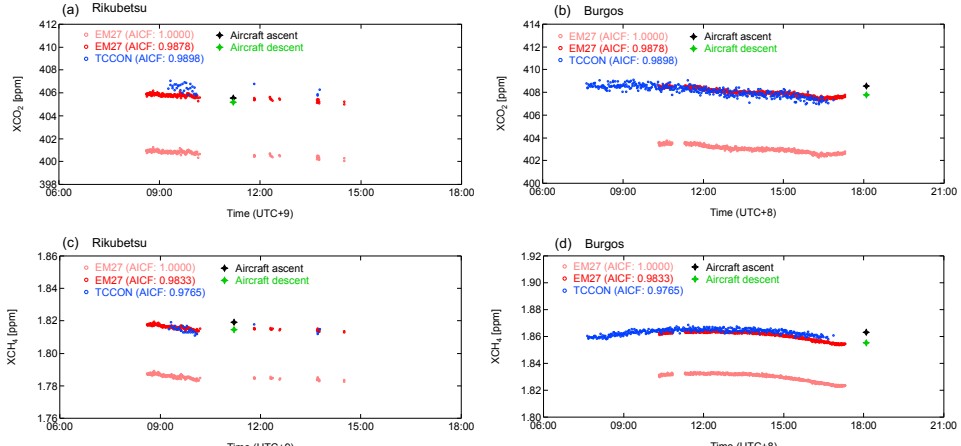

Figure 5. (a, b) $XCO_2$ and (c, d) $XCH_4$ values measured by the EM27/SUN, TCCON, and airborne instruments over (a, c) Rikubetsu on 27 April 2016 and (b, d) Burgos on 12 March 2018. The aircraft $XCO_2$ and $XCH_4$ values are calculated separately for the descending (green) and ascending (black) profiles shown in Figures 1 and 3. Shown are the EM27/SUN values without air mass independent correction factors (AICFs) and with the derived AICFs.

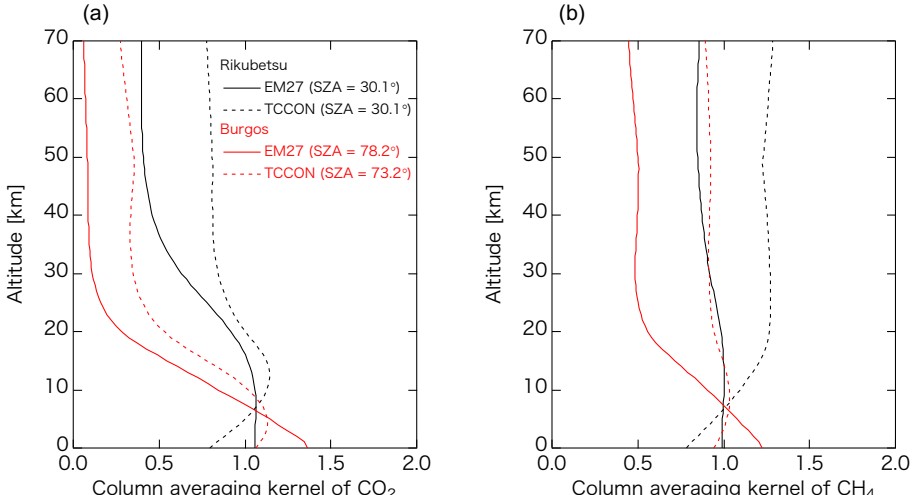

Figure 6. Column averaging kernels of (a) $CO_2$ and (b) $CH_4$ retrievals from the EM27/SUN
and TCCON spectra, which are used for calculating the aircraft $XCO_2$ and $XCH_4$ values over
Rikubetsu (black) and Burgos (red).