# Peer review of "Validation of XCO2 and XCH4 retrieved from a portable Fourier transform spectrometer with those from in-situ profiles from aircraft borne instruments"

_Atmospheric Measurement Techniques, 2020_

## Referee Comment (RC1) · Anonymous Referee #1 · 18 Jun 2020

**General comments**

The manuscript "Validation of $X_{CO_2}$ and $X_{CH_4}$ retrieved from a portable Fourier transform spectrometer with those from in-situ profiles from aircraft borne instruments" by Hirofumi Ohyama et. al. describes the validation of retrievals of the column averaged dry air mole fractions of $CO_2$ and $CH_4$ from a single portable, low-resolution near infrared solar absorption EM27/SUN Fourier transform spectrometer at the Rikubetsu and Burgos total carbon column observing network (TCCON) sites with in situ aircraft measurements.

[Figure]

The presented work represents one of the first documented examples of in situ valida-
tion of greenhouse gas measurements from a portable spectrometer of this type and
therefore contributes significantly to the value of such measurement techniques.

The Authors have taken rigorous steps to ensure the robustness of the comparisons
by demonstrating the stability of the portable instrument in terms of its instrument line
shape and comparison of retrievals to the Tsukuba TCCON site, and by choosing
which aircraft data to compare to, informed by the effect of large scale dynamics on
the tropopause height in the case of the Rikubetsu comparison and by transport of
regional emissions for Burgos.

The manuscript is well written and follows a logical narrative. All important steps are
outlined, and assumptions appropriately justified. I would strongly recommend publi-
cation of the manuscript subject to some minor alterations outlined below.

**Specific comments**

At the end of sections 3.1 and 3.2, and elsewhere in the manuscript particularly Table 2,
the terms uncertainty and error are used interchangeably. The error in a measurement
should refer to the difference between that measurement and the true value of the
measurand whereas the uncertainty describes the range about the measurement in
which the true value most likely lies. In the context of this work, the term uncertainty
should be used. For further information I refer the authors to the BIPM Guide to the
Expression of Uncertainty in Measurement.

To aid with the understanding of the choice of aircraft profile used for the Rikubetsu
comparison it would be helpful if the radiosonde lapse rate derived tropopause heights
(or a subset thereof) and the GGG derived value were plotted on Figs 1 (b) and (c) or
Fig 2 (a), and the GGG determined tropopause height included in Table 1.

Figure 1 (b) seems to be missing data from the ascent profile between just above the
surface and approximately 3 km. It would also aid the interpretation if Figures 1 and

2, (b) and (c) included an indication of the transition from aircraft data to a priori in the composite profile.

It should be made clearer that the EM27 results presented in Table 4 are before the derived airmass independent correction factor has been applied.

Has the GGG2014 airmass dependent correction factor also been applied to the EM27 retrievals presented?

Past and present tenses are used inconsistently through the manuscript, this should be rectified.

Page 6, line 195 insert CO2 before profiles when referencing figure 2 (a).

---

## Referee Comment (RC2) · Anonymous Referee #2 · 14 Jul 2020

The paper "Validation of XCO2 and XCH4 retrieved from a portable Fourier transform spectrometer with those from in-situ profiles from aircraft borne instruments" by Ohyama et al. presented the validation study of EM27/SUN using in-situ aircraft profile measurements. They have done thorough analyses of the descending and ascending aircraft profiles and used descending profile to derive the correction factors for XCO2 and XCH4 for the portable FTS. The paper is clearly written and the approach and technical details are well elaborated and presented. Because this study is one of the first attempt to derive the correction factors for EM27/SUN using in-situ profiles from aircraft

borne instruments, I recommend publication with the following comments addressed:

1. Abstract: The sentence "The EM27/SUN XCO2 and XCH4 data . . . were not applied" is in my opinion redundant. You may remove this sentence and add a separate sentence to compare the correction factors for EM27/SUN and TCCON instrument.

2. Line 84: other satellite validation studies include: https://www.atmos-meas-tech-discuss.net/amt-2020-19/ https://www.atmos-meas-tech.net/8/5023/2015/amt-8-5023-2015.pdf

3. Line 85: Long-term observations using EM27/SUN have also been conducted in urban areas, for example in Munich when deploying an automated enclosure system (https://www.atmos-meas-tech.net/11/2173/2018/amt-11-2173-2018.html).

4. The authors have specified the total time duration for the ascending and descending flights, could you please specify how long they take individually to get a sense about the time duration for the profile sampling.

5. Line 227: can you please elaborate more in detail how did you determine the errors in the aircraft CO2 on the basis of precision and accuracy of the LICOR NDIR spectrometer?

6. Figure 1 and Figure 3, subfigures b and c: Can you please show/mark the tropospheric heights together with the measurement profiles?

7. Line 354: compare the influence of the transport on XCO2 and XCH4 with the uncertainties. Please specify the uncertainties or referring a citation e.g. Frey et al. 2019.

8. Line 377: the number of EM27/SUN data are 4 and 24 according to the temporal coincident criterion are not really visible in the Figure 5.

9. I would include the instrument line function parameters in the table 3 or table 4. The different instrument line function of the EM27/SUN at different locations could be part

of the reason for the different relative differences.

10. Figure 5: it is hard to see the comparison between corrected EM27/SUN, TCCON and airborne instruments, maybe you can zoom in a bit.

11. Figure 5, caption: without …. (AICFs = 1) and with (AICFs $\sim$= 1)

12. Table 3 caption: we note -> please note

13. I would recommend language check including usage for commas and consistency check for past and present tenses.

---

## Author Response (AR1)

**Point-by-point responses to the reviewers' comments.**

**Reviewer #1**

**General comments**

The manuscript "Validation of XCO2 and XCH4 retrieved from a portable Fourier trans-form spectrometer with those from in-situ profiles from aircraft borne instruments" by Hirofumi Ohyama et al. describes the validation of retrievals of the column averaged dry air mole fractions of CO2 and CH4 from a single portable, low-resolution near infrared solar absorption EM27/SUN Fourier transform spectrometer at the Rikubetsu and Burgos total carbon column observing network (TCCON) sites with in situ aircraft measurements.

The presented work represents one of the first documented examples of in situ validation of greenhouse gas measurements from a portable spectrometer of this type and therefore contributes significantly to the value of such measurement techniques. The Authors have taken rigorous steps to ensure the robustness of the comparisons by demonstrating the stability of the portable instrument in terms of its instrument line shape and comparison of retrievals to the Tsukuba TCCON site, and by choosing which aircraft data to compare to, informed by the effect of large scale dynamics on the tropopause height in the case of the Rikubetsu comparison and by transport of regional emissions for Burgos. The manuscript is well written and follows a logical narrative. All important steps are outlined, and assumptions appropriately justified. I would strongly recommend publication of the manuscript subject to some minor alterations outlined below.

We thank you for reading our paper carefully and providing valuable comments. We have added some descriptions for clarification and revised our manuscript according to your comments. Please see our specific responses below.

**Specific comments**

At the end of sections 3.1 and 3.2, and elsewhere in the manuscript particularly Table 2, the terms uncertainty and error are used interchangeably. The error in a measurement should refer to the difference between that measurement and the true value of the measurand whereas the uncertainty describes the range about the measurement in which the true value most likely lies. In the context of this work, the term uncertainty should

be used. For further information I refer the authors to the BIPM Guide to the Expression of Uncertainty in Measurement.

We have revised the text to exclusively use the term uncertainty, unifying the two terms (i.e., uncertainty and error).

To aid with the understanding of the choice of aircraft profile used for the Rikubetsu comparison it would be helpful if the radiosonde lapse rate derived tropopause heights (or a subset thereof) and the GGG derived value were plotted on Figs 1 (b) and (c) or Fig 2 (a), and the GGG determined tropopause height included in Table 1.

We have added the tropopause heights from the radiosonde lapse rate and the GGG2014 in Figs. 1b and 1c. In addition, the tropopause height from the GGG2014 has been included in Table 1.

Figure 1 (b) seems to be missing data from the ascent profile between just above the surface and approximately 3 km. It would also aid the interpretation if Figures 1 and 2, (b) and (c) included an indication of the transition from aircraft data to a priori in the composite profile.

As you pointed out, there is no description of the missing data from the ascent profile. We have added the following sentence in Sect. 3.1 (lines 190–191): "There are missing data due to instrumental calibrations, especially between 0.24 and 2.78 km of the CO2 ascent profile (Fig. 1b)." Additionally, we have added the following sentences in Sect. 3.4 (lines 412–416): "When calculating aircraft XCO2 and XCH4 values, the missing data were linearly interpolated. We note that, provided that the missing data between 0.24 and 2.78 km of the CO2 ascent profile were substituted by the descent profile in the corresponding altitude range, the difference between the XCO2 values from the linear interpolation and the substitution was less than 0.1 ppm."

Regarding the transition from aircraft data to the a priori profile, we have added the composite profiles in Figs. 1b, 1c, 3b, and 3c.

It should be made clearer that the EM27 results presented in Table 4 are before the derived airmass independent correction factor has been applied.

We have added the following sentence in the caption of Table 4: "The air mass independent correction factors derived in this study are not yet applied to the EM27/SUN data."

Has the GGG2014 airmass dependent correction factor also been applied to the EM27 retrievals presented?

Yes. We have revised the description related to the correction factors in Sect. 2.1 (lines 143–148) as follows: "The GGG2014 software includes air mass independent and air mass dependent correction factors for the TCCON data. The air mass independent correction factors (AICFs) were not utilized (i.e., they were set to one) because we separately determined them for EM27/SUN in this study. Meanwhile, we used the same air mass dependent correction factors (ADCFs) as those applied to the TCCON data, and their validity is evaluated in Sect. 3.3."

In addition, we have added the following sentences in Sect. 3.3 (lines 370-386): "As described in Sect. 2.1, we applied the GGG2014 ADCFs to the EM27/SUN retrievals. The ADCF is a coefficient tied to a symmetric basis function (Eq. A12 in Wunch et al. (2011a)) representing spurious diurnal variation, and the values derived from the TCCON data at multiple sites are  $-0.0068 \pm 0.0050$  for XCO2 and  $0.0053 \pm 0.0080$  for XCH4 (Wunch et al., 2015). To assess the relevance of applying the ADCFs derived from the TCCON data to the EM27/SUN data, we derived the ADCF for our EM27/SUN, such that the difference between the EM27/SUN and TCCON retrievals in Burgos that were individually averaged into 10 min bins is minimized while taking into account a coefficient for correcting the mean bias between EM27/SUN and the TCCON data. The derived ADCFs are  $-0.0064 \pm 0.0004$  for XCO2 and  $0.0034 \pm 0.0007$  for XCH4 (the uncertainties were estimated as  $1\sigma$  standard deviations of daily ADCFs derived from four days side by side observations in Burgos). The ADCFs for XCO2 show good agreement between the EM27/SUN and the TCCON, while those for XCH4 show a slightly larger difference. Considering that the ADCFs for our instrument are consistent with those for the TCCON data within the uncertainties and that the ADCFs have the possibility to vary with the seasons and sites (Wunch et al., 2015), we conclude that the use of the mean ADCFs derived from the TCCON data is a reasonable choice."

**References:**

Wunch, D., Toon, G. C., Blavier, J. F., Washenfelder, R. A., Notholt, J., Connor, B. J., Griffith, D. W., Sherlock, V., and Wennberg, P. O.: The total carbon column observing network, Philos. Trans. A Math. Phys. Eng. Sci., 369, 2087–2112, https://doi.org/10.1098/rsta.2010.0240, 2011a.

Wunch, D., Toon, G. C., Sherlock, V., Deutscher, N. M., Liu, C., Feist, D. G., and Wennberg, P. O.: The Total Carbon Column Observing Network's GGG2014 Data Version, Tech. rep., California Institute of Technology, Pasadena, CA, https://doi.org/10.14291/tccon.ggg2014.documentation.R0/1221662, 2015.

Past and present tenses are used inconsistently through the manuscript, this should be rectified.

We have revised the manuscript based on the following basis. We used the past tense to describe measurements and analyses that have already been completed at the time of writing of the paper, while we used the present tense to interpret the results and discuss the significance of the findings.

Page 6, line 195 insert CO2 before profiles when referencing figure 2 (a). We have revised accordingly.

**Reviewer #2**

The paper "Validation of XCO2 and XCH4 retrieved from a portable Fourier transform spectrometer with those from in-situ profiles from aircraft borne instruments" by Ohyama et al. presented the validation study of EM27/SUN using in-situ aircraft profile measurements. They have done thorough analyses of the descending and ascending aircraft profiles and used descending profile to derive the correction factors for XCO2 and XCH4 for the portable FTS. The paper is clearly written and the approach and technical details are well elaborated and presented. Because this study is one of the first attempt to derive the correction factors for EM27/SUN using in-situ profiles from aircraft borne instruments, I recommend publication with the following comments addressed:

We thank you for reading our paper carefully and providing valuable comments. We have revised our manuscript according to your comments. Please see our specific responses below.

1. Abstract: The sentence "The EM27/SUN XCO2 and XCH4 data...were not applied" is in my opinion redundant. You may remove this sentence and add a separate sentence to compare the correction factors for EM27/SUN and TCCON instrument.

We have simplified the sentence as follows: "The EM27/SUN XCO2 and XCH4 data were derived by using the GGG2014 software without applying air mass independent correction factors (AICFs)." In addition, we have added the following sentence (lines 55–56): "Applying AICFs being utilized for the TCCON data (0.9898 for XCO2 and 0.9765 for XCH4) to the EM27/SUN data induces an underestimate for XCO2 and an overestimate for XCH4."

2. Line 84: other satellite validation studies include: https://www.atmos-meas-tech-discuss.net/amt-2020-19/

https://www.atmos-meas-tech.net/8/5023/2015/amt-8-5023-2015.pdf

We have revised the sentence as follows: "An additional observation campaign for satellite data validation was conducted in the desert areas of Australia (Velazco et al., 2019). Furthermore, EM27/SUN data obtained above the Atlantic Ocean (Klappenbach et al., 2015) and in boreal areas (Tu et al., 2020) have been utilized for satellite validation studies."

3. Line 85: Long-term observations using EM27/SUN have also been conducted in urban areas, for example in Munich when deploying an automated enclosure system (https://www.atmos-meas-tech.net/11/2173/2018/amt-11-2173-2018.html).

We have revised the sentence as follows: "Long-term observations have also been conducted in Africa where operational observation by the IFS 125HR is difficult (Frey et al., 2020), and in urban areas, e.g. in Munich when deploying an automated enclosure system (Heinle and Chen, 2018)."

4. The authors have specified the total time duration for the ascending and descending flights, could you please specify how long they take individually to get a sense about the time duration for the profile sampling.

We have revised the sentences on the KORUS-AQ flight as follows: "The descending profile was measured from 10.81 to 0.10 km *in*  $\sim$ 34 *min* with a spiral flight pattern over the Rikubetsu site. The ascending profile was measured up to an altitude of 11.51 km *in*  $\sim$ 27 *min* in a linear manner on the west side of the Rikubetsu site."

Similarly, the sentences on the EMeRGe flight have been revised as follows: "The descending profile was measured from 6.47 to approximately 0.6 km *in* ~20 *min* approaching the Burgos site from south to northeast. The low-level flight at approximately 0.6 km was performed as near as possible to the north side of the Burgos site *for* ~9 *min*. The ascending profile was measured up to 9.32 km *in* ~11 *min* after the low-level flight west of the Burgos site. Additional data for the profiles above 6.47 (descent flight) and 9.32 km (ascent flight) were taken from the same aircraft data measured during the descent flight *lasting for* ~10 *min* from an altitude of 13.87 km west of Manila."

5. Line 227: can you please elaborate more in detail how did you determine the errors in the aircraft  $CO_2$  on the basis of precision and accuracy of the LICOR NDIR spectrometer?

We have added the following sentence in Sect. 3.1 (lines 239-241): "We estimated the uncertainties in the aircraft CO2 data to be 0.27 ppm from the square root of the sum of the squares of both a precision of 0.1 ppm and an accuracy of 0.25 ppm (Vay et al., 2011; Tang et al., 2018)."

6. Figure 1 and Figure 3, subfigures b and c: Can you please show/mark the tropospheric heights together with the measurement profiles?

We have added three types of tropopause heights (lapse rate tropopause, dynamical tropopause, and the GGG2014 derived tropopause) in Figs. 1b and 1c and have added the GGG2014 derived tropopause in Figs. 3b and 3c.

7. Line 354: compare the influence of the transport on  $XCO_2$  and  $XCH_4$  with the uncertainties. Please specify the uncertainties or referring a citation e.g. Frey et al. 2019. We have revised the sentence as follows: "Thus, the influence of EM27/SUN transports on the  $XCO_2$  and  $XCH_4$  retrievals are comparable to their  $2\sigma$  uncertainties (0.6 ppm for  $XCO_2$  and 2.2 ppb for  $XCH_4$  (Frey et al., 2019))."

8. Line 377: the number of EM27/SUN data are 4 and 24 according to the temporal coincident criterion are not really visible in the Figure 5.

In Figure 5, we have highlighted the time satisfying the temporal coincident criterion.

9. I would include the instrument line function parameters in the table 3 or table 4. The different instrument line function of the EM27/SUN at different locations could be part of the reason for the different relative differences.

We have added the modulation efficiency in Table 3 and have added the following sentences in Sect. 3.4 (lines 429–435): "Provided that the mean value of the modulation efficiency before and after the transport was that during the campaign, the difference in the modulation efficiency between the campaigns (EMeRGe – KORUS-AQ) was – 0.0031 (Table 3), which corresponds to a change of -0.047 % for the XCO2 value. Because the relative difference between the EM27/SUN and the aircraft XCO2 data differed by -0.072 % (Table 4) between the campaigns (EMeRGe – KORUS-AQ), the change in the ILS of the EM27/SUN for the campaign periods may have partly contributed to the difference in the relative differences."

10. Figure 5: it is hard to see the comparison between corrected EM27/SUN, TCCON and airborne instruments, maybe you can zoom in a bit.We have changed the scale of y-axis of Figure 5.

11. Figure 5, caption: without.... (AICFs = 1) and with (AICFs  $\sim$ = 1) We have revised the caption.

12. Table 3 caption: we note -> please noteWe have revised the caption.

13. I would recommend language check including usage for commas and consistency check for past and present tenses.

A language check has been conducted for the entire manuscript by a native English speaker who is a co-author.

**A list of all relevant changes made in the manuscript except those described in the responses to the reviewer comments.**

• We have corrected the last name of Charles C.-K. Chou and have added the middle name of Nicholas M. Deutscher.

• The aircraft XCO2 and XCH4 data have slightly changed (Tables 2, 4, and 5) because of a minor bug in the program that calculates the XCO2 and XCH4 values.

• We have added author contributions.

**Validation of XCO2 and XCH4 retrieved from a portable Fourier transform spectrometer with those from in-situ profiles from aircraft borne instruments**

Hirofumi Ohyama1, Isamu Morino1, Voltaire A. Velazco2,3, Theresa Klausner4, Gerry Bagtasa5, Matthäus Kiel6, Matthias Frey1, Akihiro Hori1, Osamu Uchino1, Tsuneo Matsunaga1, Nicholas M. Deutscher2, Joshua P. DiGangi7, Yonghoon Choi7, Glenn S. Diskin7, Sally E. Pusede8, Alina Fiehn4, Anke Roiger4, Michael Lichtenstern4, Hans Schlager4, Pao K. Wang9, Charles C.-K. Chou9, Maria Dolores Andrés-Hernández10, and John P. Burrows10

1 National Institute for Environmental Studies, Tsukuba, Japan

2 Centre for Atmospheric Chemistry, University of Wollongong, NSW 2522, Australia

3 Oscar M. Lopez Center for Climate Change Adaptation and Disaster Risk Mgmt. Foundation Inc., Manila, Philippines

4 Deutsches Zentrum für Luft- und Raumfahrt (DLR), Institut für Physik der Atmosphäre, Oberpfaffenhofen, Germany

5 Institute of Environmental Science & Meteorology, University of the Philippines, Diliman Quezon City, Philippines

6NASA Jet Propulsion Laboratory, California Institute of Technology, Pasadena, CA, USA

7 NASA Langley Research Center, Hampton, VA, USA

8 Department of Environmental Sciences, University of Virginia, Charlottesville, VA, USA

9 Research Center for Environmental Changes, Academia Sinica, Taipei, Taiwan

10 Institute of Environmental Physics, University of Bremen, Otto-Hahn-Allee 1, 28359 Bremen, Germany

**Abstract**

Column-averaged dry-air mole fractions of carbon dioxide (XCO2) and methane (XCH4) measured by a solar viewing portable Fourier transform spectrometer (FTS, EM27/SUN) have been characterized and validated by comparison using in-situ profile measurements made during the transfer flights of two aircraft campaigns: Korea-United States Air Quality Study (KORUS-AQ) and Effect of Megacities on the Transport and Transformation of Pollutants on the Regional and Global Scale (EMeRGe). The aircraft flew over two Total Carbon Column Observing Network (TCCON) sites: Rikubetsu, Japan (43.46° N, 143.77° E) for the KORUS-AQ campaign and Burgos, Philippines (18.53° N, 120.65° E) for the EMeRGe campaign. The EM27/SUN was deployed at the corresponding TCCON sites during the overflights. The mole fraction profiles obtained by the aircraft over Rikubetsu differed between the ascending and the descending flights above approximately 8 km for both CO2 and CH4. Because the spatial pattern of tropopause heights based on potential vorticity values from the ERA5 reanalysis shows that the tropopause height over the Rikubetsu site was consistent with the descending profile, we used only the descending profile to compare with the EM27/SUN data. Both the XCO2 and XCH4 derived from the descending profiles over Burgos were lower than those from the ascending profiles. Output from the Weather Research and Forecast Model indicates that higher CO2 for the ascending profile originated in central Luzon, an industrialized and densely populated region about 400 km south of the Burgos TCCON site. Air masses observed with the EM27/SUN overlap better with those from the descending aircraft profiles than those from the ascending aircraft profiles with respect to their properties such as origin and atmospheric residence times. Consequently, the descending aircraft profiles were used for the comparison with the EM27/SUN data. The EM27/SUN XCO2 and XCH4 data were derived by using the GGG2014 software without applying air mass independent correction factors (AICFs), The comparison of the EM27/SUN observations with the aircraft data revealed that on average, the EM27/SUN XCO2 data were biased low by 1.22 % and the EM27/SUN XCH4 data were biased low by 1.71 %. The resulting AICFs of 0.9878 for XCO2 and 0.9829 for XCH4 were obtained for the EM27/SUN. Applying AICFs being utilized for the TCCON data (0.9898 for XCO2 and 0.9765 for XCH4) to the EM27/SUN data induces an underestimate for XCO2 and an overestimate for XCH4.

[revised manuscript text omitted]

(after Burgos obs.)

Table 4. Comparison of EM27/SUN data with aircraft  $XCO_2$  and  $XCH_4$  data. The air mass independent correction factors derived in this study are not yet applied to the EM27/SUN data. The relative differences are calculated as follows: (EM27/SUN – Aircraft) / Aircraft × 100.

| т.,:       |                  | $\rm XCO_2$ |                | $\rm XCH_4$ |               |                |  |
|------------|------------------|-------------|----------------|-------------|---------------|----------------|--|
| Location   | EM27/SUN Aircraf |             | Relative       | EM27/SUN    | Aircraft      | Relative       |  |
| (Campaign) | [ppm]            | [ppm]       | difference [%] | [ppb]       | [ppb]         | difference [%] |  |
| Rikubetsu  | 400.40           | 405.27      | 1 170          | 1794.9      | 1914 6        | 1.642          |  |
| (KORUS-AQ) | 400.49           | 405.27      | -1.179         | 1/84.8      | 1814.0 | -1.042         |  |
| Burgos     | 402.64           | 407.74      | -1 251         | 1822 /      | 1856.2        | _1 772         |  |
| (EMeRGe)   | 402.04           | +07.74      | -1.231         | 1023.4      | 1030.3        | -1.//2  |  |

| 削除: -        | -1.172                                          |                                                                                                                                                                            |                                                                                                                                                                            |                                                                                                                                                 |                                                                                                                                                 |                                                                                                                                                 |
|--------------|-------------------------------------------------|----------------------------------------------------------------------------------------------------------------------------------------------------------------------------|----------------------------------------------------------------------------------------------------------------------------------------------------------------------------|-------------------------------------------------------------------------------------------------------------------------------------------------|-------------------------------------------------------------------------------------------------------------------------------------------------|-------------------------------------------------------------------------------------------------------------------------------------------------|
| 削除: -        | -1.572                                          |                                                                                                                                                                            |                                                                                                                                                                            |                                                                                                                                                 |                                                                                                                                                 |                                                                                                                                                 |
| 削除: 4 | 05.24                                           |                                                                                                                                                                            |                                                                                                                                                                            |                                                                                                                                                 |                                                                                                                                                 |                                                                                                                                                 |
| 削除: 1 | 813.3                                           |                                                                                                                                                                            |                                                                                                                                                                            |                                                                                                                                                 |                                                                                                                                                 |                                                                                                                                                 |
| 削除: -        | -1.260                                          |                                                                                                                                                                            |                                                                                                                                                                            |                                                                                                                                                 |                                                                                                                                                 |                                                                                                                                                 |
| 削除: -        | -1.772                                          |                                                                                                                                                                            |                                                                                                                                                                            |                                                                                                                                                 |                                                                                                                                                 |                                                                                                                                                 |
| 削除: 4 | 07.78                                           |                                                                                                                                                                            |                                                                                                                                                                            |                                                                                                                                                 |                                                                                                                                                 |                                                                                                                                                 |
| 削除: 1 | 856.3                                           |                                                                                                                                                                            |                                                                                                                                                                            |                                                                                                                                                 |                                                                                                                                                 |                                                                                                                                                 |
|              | 削除: - 削除: - 削除: 4 削除: 1 削除: - 削除: - 削除: 4 削除: 1 |  <li>制除: -1.172</li> <li>削除: -1.572</li> <li>削除: 405.24</li> <li>削除: 1813.3</li> <li>削除: -1.260</li> <li>削除: -1.772</li> <li>削除: 407.78</li> <li>削除: 1856.3</li>  |  <li>削除: -1.172</li> <li>削除: -1.572</li> <li>削除: 405.24</li> <li>削除: 1813.3</li> <li>削除: -1.260</li> <li>削除: -1.772</li> <li>削除: 407.78</li> <li>削除: 1856.3</li>  | 削除: -1.172 削除: -1.572 削除: 405.24 削除: 1813.3 削除: -1.260 削除: -1.772 削除: 407.78 削除: 1856.3 | 制除: -1.172 制除: -1.572 制除: 405.24 制除: 1813.3 制除: -1.260 制除: -1.772 制除: 407.78 制除: 1856.3 | 削除: -1.172 削除: -1.572 削除: 405.24 削除: 1813.3 削除: -1.260 削除: -1.772 削除: 407.78 削除: 1856.3 |

|                          | XCO 2 |               |                | XCH 4 |               |                |
|--------------------------|------------------|---------------|----------------|------------------|---------------|----------------|
| Location -
(Campaign) | TCCON            | Aircraft      | Relative       | TCCON            | Aircraft      | Relative       |
|                          | [ppm]            | [ppm]         | difference [%] | [ppb]            | [ppb]         | difference [%] |
| Rikubetsu                | 406.45           | 404.02        | 0.275          | 1914 7           | 1810.5        | 0.222          |
| (KORUS-AQ)               |                  | 404.93        | 0.373   | 1014./           | 1810.5 | 0.232          |
| Burgos                   | 407.53           | 407.64 | -0.027_        | 1855.5           | 1854.4        | D .059  |
| (EMeRGe)                 |                  | ¥             | <del>-</del>   |                  | Terri ferri i | -              |

Table 5. Comparison of TCCON data with aircraft  $XCO_2$  and  $XCH_4$  data. The relative differences are calculated as follows: (TCCON – Aircraft) / Aircraft × 100.

|   | 削除: | 404.97 |
|---|-----|--------|
|   | 削除: | 0.365  |
|   | 削除: | 1809.8 |
|   | 削除: | 0.271  |
|   | 削除: | -0.061 |
|   | 削除: | 407.78 |
|   | 削除: | 1855.4 |
| ľ | 削除: | 0.005  |